# Use of lay vaccinators in animal vaccination programmes: A scoping review

Christian Tetteh Duamor [1,2]*, Katie Hampson [3], Felix Lankester [4,5], Maganga Sambo [2], Katharina Kreppel [1], Sally Wyke [6], Sarah Cleaveland [3]

1 Department of Global Health, Nelson Mandela African Institution of Science and Technology, Arusha, Tanzania, 2 Environmental Health and Ecological Sciences Thematic Group, Ifakara Health Institute, Ifakara, Tanzania, 3 Institute of Biodiversity, Animal Health & Comparative Medicine, College of Medical, Veterinary and Life Sciences, University of Glasgow, Glasgow, United Kingdom, 4 Paul G. Allen School for Global Health, Washington State University, Pullman, Washington, United States of America, 5 Global Animal Health Tanzania, Arusha, Tanzania, 6 Institute of Health and Wellbeing, College of Social Sciences, University of Glasgow, Glasgow, United Kingdom

* duamort@gmail.com

**Data Availability Statement:** All relevant data are within the manuscript.

**Funding:** This study was supported by the following funds. CTD, SC, KK: Funding for

## Abstract

### Background

The human resource gap in veterinary sectors, particularly in low-income countries, imposes limitations on the delivery of animal healthcare in hard-to-reach populations. Lay animal health workers have been deployed in these settings to fill the gap though there are mixed views about the benefits of doing this and whether they can deliver services safely. We mapped evidence on the nature and extent of roles assigned to lay animal vaccinators, and identified lessons useful for their future deployment.

### Methodology/Principal findings

Following the PRISMA Extension for Scoping Reviews guidelines, we searched seven bibliographic databases for articles published between 1980 and 2021, with the search terms lay OR community-based OR volunteer AND "animal health worker" OR vaccinator*, and applied an *a priori* exclusion criteria to select studies. From 30 identified studies, lay vaccinators were used by non-government developmental (n = 12, 40%), research (n = 10, 33%) and government (n = 5, 17%) programmes to vaccinate domestic animals. The main reason for using lay vaccinators was to provide access to animal vaccination in the absence of professional veterinarians (n = 12, 40%). Reported positive outcomes of programmes included increased flock and herd sizes and farmer knowledge of best practice (n = 13, 43%); decreased disease transmission, outbreaks and mortality (n = 11, 37%); higher vaccination coverage (10, 33%); non-inferior seroconversion and birth rates among vaccinated herds (n = 3, 10%). The most frequently reported facilitating factor of lay vaccinator programmes was community participation (n = 14, 47%), whilst opposition from professional veterinarians (n = 8, 27%), stakeholders seeking financial gains to detriment of programmes goals (n = 8, 27%) and programming issues (n = 8, 27%) were the most frequently reported barriers. No study reported on cost-effectiveness and we found no record from a low and middle-income country of lay vaccinator programmes being integrated into national veterinary services.

postgraduate study (CTD) and preparation of the article was received from the DELTAS Africa Initiative [Afrique One-ASPIRE /DEL-15-008]. Afrique One-ASPIRE is funded by a consortium of donors, including the African Academy of Sciences (AAS), Alliance for Accelerating Excellence in Science in Africa (AESA), the New Partnership for Africa's Development Planning and Coordinating (NEPAD) Agency, the Wellcome Trust [107753/A/15/Z] and the UK government. CTD received monthly stipend from ASPIRE. FL, SC, KH, SW: Collaborative work for this review was funded through the National Institutes of Health [R01AI141712] and MSD Animal Health. This review was conducted as a recommendation from a meeting funded by the University of Glasgow Small Grants Fund, supported by an allocation of GCRF funding administered by the Scottish Funding Council. K.H. was funded by the Wellcome Trust [207569/Z/17/Z]. The funders had no role in study design, data collection and analysis, decision to publish, or preparation of the manuscript.

**Competing interests:** The authors have declared that no competing interests exist.

## Conclusion

Although the majority of included studies reported more benefits and positive perceptions of lay vaccinator programmes than problems and challenges, regularization will ensure the programmes can be designed and implemented to meet the needs of all stakeholders.

## Author summary

In the absence of professional animal healthcare workers in hard-to-reach settings, lay persons, with limited, non-formal training, have been used to provide animal healthcare services, including vaccination. In spite of the perceived crucial roles lay persons play in the animal health sector, their services are largely unrecognized within official animal healthcare systems. We compiled evidence on how lay persons have been used in animal vaccination programmes and make recommendations regarding how they can be used in more effective ways. We found they were used by both government and non-government institutions to vaccinate different domestic animals and provide regular animal healthcare services. They were mainly used where professional animal healthcare workers are not available or are limited in number. The programmes were more successful where they had the support of the public and institutions, and their outcomes were largely similar to those delivered by professionals. We also found that community participation was an important facilitating factor, whilst the main challenges they faced were opposition from professional veterinarians, financial interests of stakeholders and planning issues. We concluded that lay animal vaccinator programmes could be more beneficial if better regulated.

## Introduction

Health interventions have, in many instances, relied upon nonprofessional personnel as a stop-gap measure to deliver essential services in settings where there is limited availability of professionals. In human health, for example, nonprofessional health workers have made critical contributions to the large-scale delivery of human chemotherapeutic programmes to control diseases such as schistosomiasis, lymphatic filariasis and onchocerciasis, to the testing and treating of uncomplicated malaria, to the distribution of insecticide treated bed nets and vitamin-A supplements [1–5], and to vaccination campaigns aiming to eradicate polio [6]. Indeed, the eradication of smallpox was made possible through the participation of nonmedical personnel in the community-wide vaccination campaigns needed to achieve herd immunity [7]. In England, volunteers are currently being recruited from a range of professions for mass vaccination against Covid-19 [8,9] with scaling up of vaccination facilitated by amendments to regulations that allow for temporary authorizations and expansion of the workforce who can administer vaccines [10]. In the same vein, lay animal health workers have also been deployed in the animal health sector, and were key to the success of mass cattle vaccination campaigns leading to the eradication of rinderpest [11].

The World Health Organization defines nonprofessional or lay health workers, also known as village or community health workers, as health workers who are given limited, non-formal professional training to perform health care delivery functions in the context of an intervention [12,13]. In animal health care, lay workers, also known as community (based) animal health workers (CAHWs), have a different status from that of veterinary paraprofessionals, livestock field officers or animal health technicians who have undergone an officially

recognized training and are formally integrated within veterinary systems [14]. During interventions, CAHWs have been temporarily recruited from other professions, such as the army, police or environmental health officers, nurses, teachers, or even be retirees, farmers or community-based volunteers [14,15]. While the contribution of paraprofessionals is formally recognized by the World Animal Health Organization (OIE) [16], the effective deployment of CAHWs still faces major challenges.

The concept of Community-Based Animal Health Workers and Animal Health Assistants gained traction in the 1970s with the World Bank advocating that livestock producers' associations should include grassroots-level animal health workers [17]. The concept was developed further in the 1990s following the structural adjustment programmes of the 1970s and 1980s which required many developing economies to embark on trade liberalization, deregulation and privatization of certain public services. From the perspective of veterinary services, these programmes compelled governments to rely on private veterinary service providers in hard-to-reach communities [18,19]. For example, in 1988, the Kenyan Government stopped automatic employment of graduating veterinarians and animal health technicians, who had previously been deployed to replace a cadre of workers known as vet scouts, and were providing local veterinary services free-of-charge [20]. Instead, graduates were encouraged into private practice. However, the establishment of private professional veterinary care in rural and/ or remote settings was met with several challenges such as the inability of farmers in these areas to afford their services, lack of infrastructure to support their work and the preference of most veterinarians to stay in urban centers that were inaccessible to farmers in rural areas [21–23]. In responding to these challenges, international and non-governmental organizations began to champion the concept of CAHWs for delivery of animal health services in remote, least developed and conflict-stricken settings where livestock constitute valuable economic and social assets for households [24]

Earlier reviews [25,26] indicated that CAHW programmes have been implemented widely around the world and there is renewed interest in their potential to improve access to animal health care service delivery and disease surveillance in resource limited settings. CAHW programmes are also recognized as having clear potential to contribute to the progress of many of the Sustainable Development Goals [27]. However, these reviews also highlight several regulatory, social and sustainability challenges, including, misconduct on the part of lay animal health workers, loss of indigenous veterinary knowledge and, when farmers had to pay for animal health care services, the redirection of scarce resources from the welfare of women and girls [25,26].

Animal vaccination underpins the prevention and control of many major animal diseases, including zoonoses, and is an area of animal health where the contribution of CAHWs has been widely advocated to support global elimination strategies for diseases such as peste des petits ruminants (PPR), foot and mouth disease and rabies [28–30]. In the case of rabies, the limited availability of professional and paraprofessional veterinarians across the majority of rabies endemic countries means that lay animal vaccinators could be a critical human resource to support scaled up mass dog vaccination campaigns towards achieving the global goal of zero human deaths from rabies by 2030 [1,31]. The discovery of thermostable properties of the Nobivac rabies vaccine [32] and the feasibility of storing these vaccines in locally made cooling devices [33] makes it possible for these vaccines to be stored and used in remote settings by trained community volunteers. In spite of these possibilities to expand animal health service delivery, lay vaccinator programmes have not been formally deployed and in many countries their use is discouraged by professional veterinarians [11,34,35].

In this scoping review, we aim to map the available evidence on the nature and extent of deployment of lay vaccinators in animal vaccination programmes, the effectiveness of lay

vaccinators in delivering interventions, the factors facilitating, and the challenges associated with their use, and the resultant costs and benefits. Further, we aim to identify whether the use of lay persons for animal vaccination impacted programmatic costs and whether steps are being taken towards making the use of lay vaccinators more common.

## Methods

We used the five-stage scoping review approach [36] (identify research question, identify studies, select studies, chart data and present the findings), taking into account recent recommendations for each stage [37].

### Research questions related to the aims of the review:

i. What is the nature and extent of use of lay persons in animal vaccination programmes?

ii. How effective has the use of lay persons been in animal vaccination programmes?

iii. How has the use of lay persons for animal vaccination impacted programmatic costs?

iv. What has facilitated or hindered the use of lay persons in animal vaccination programmes?

### Study identification

We developed search terms in line with the research questions and used them to conduct an initial limited search, after which the search terms were refined. We then combined them into a standardized form as follows: lay OR community-based OR volunteer AND "animal health worker" OR vaccinator*, which we used to conduct searches in PubMed, Scopus (Elsevier), Medline, Centre for Agriculture and Biosciences International, Web of Science Core Collections, BIOSIS and Google Scholar bibliographic databases. We also manually searched electronic journals and resources including the African Journal online, Biomed Central, PLOS NTDs, ResearchGate and ScienceDirect, as well as reference lists from authors and the included studies. The first search was conducted in December, 2020 and repeated in March, 2021 and covered studies published since 1980.

### Study selection

Two reviewers, CTD and MS, independently screened titles, abstracts and full texts and selected studies based on *a priori* inclusion/exclusion criteria. Studies were included if: i) they were peer-reviewed, ii) published in English language and iii) the title or abstract referred to or described implementation of animal vaccination AND referred to either lay vaccinators, community animal health workers, volunteer vaccinators, community-based animal health workers or vaccinators. Where there were disagreements, CTD and MS met to resolve them, and studies that could not be decided on by both were reviewed by a third reviewer, SC. Corresponding authors of studies that described animal vaccination programmes but which did not report qualifications of personnel involved were contacted to confirm their qualifications and roles played.

### Data extraction

Data were extracted by CTD and MS and reviewed by SC. Information extracted for each included study were: author of study, year of publication, country of study, setting of study, objective of the study, study design employed and data collection methods. Other information included details of the nature and extent of deployment of lay animal vaccinator programmes;

evidence of their effectiveness and other benefits; facilitating factors and challenges faced. We also extracted and compiled statements relating to the facilitating factors and challenges, coded and thematically analyzed them in NVivo 12 Plus (QSR International), and tabulated the summaries. We narrate a summary of these themes across studies, presenting results in relation to the nature of the programme (research question 1), their effectiveness and other benefits (research questions 2 and 3), and facilitating factors and challenges faced (research question 4).

## Results

### Search results

Our literature searches yielded a total of 453 studies, with 321 studies remaining after duplicates were removed. Nearly half (138; 43%) of these studies were excluded at the title and abstract screening stages because they did not describe use of lay persons in the implementation of an animal vaccination programme. After full-text assessment, 27 studies were considered eligible for inclusion, having specifically described the roles of lay persons in an animal vaccination programme. Three additional studies were found from the reference lists of included papers, taking the total number of included studies to 30. Three of these were reviews and 27 were primary studies. The steps of selecting studies followed the Preferred Reporting Items for Systematic Reviews and Meta-Analyses–Extension for Scoping Reviews (PRISMA-ScR) as shown in the flow chart (Fig 1).

### Characteristics of reviewed studies

The majority of included studies, for which the dates were reported (17, 57%), were conducted after the year 2000. Most of the studies (20, 67%) were carried out in countries in eastern Africa (Ethiopia, Kenya, Malawi, Mozambique, Uganda and Tanzania), with one study each from Afghanistan, Brazil, Canada, Ghana, India, Nepal, UK, USA, South Africa and South Sudan. The studies were mainly conducted in pastoral and agropastoral settings (25, 83%), with a minority conducted in urban or suburban (4, 13%) settings. Half of the studies focused on assessing the outcome, impact or performance of CAHW programmes (15, 50%), whilst just under half focused on identifying determinants of uptake of such programmes (11, 37%). About a quarter investigated the feasibility of using this cadre of animal health workers to implement animal vaccination programmes (7, 23%). The study designs employed included case studies (3, 10%), cross-sectional surveys (12, 40%), case control study (1, 3%), experimental or randomized controlled trials (5, 20%), before-after studies (3, 10%), prospective studies (1, 3%) and reviews (5, 17%). The methods of data collection varied, but included structured or semi-structured surveys (18, 60%), ministry or programme reports (17, 57%), literature reviews (6, 20%), qualitative interviews (6, 20%), laboratory reports (6, 20%), participatory approaches (5, 17%) and non-participant observations (1, 3%). Detailed characteristics of the studies are shown in Tables 1 and 2.

### Nature of use of lay persons in animal vaccination programmes

Eleven of the projects were implemented before the year 2000 and 12 after 2000, with the years of implementation not specified for seven studies. The lay vaccinators were mainly deployed by research projects (10, 33%) or non-governmental organizational projects (12, 40%), with only five (5, 17%) being government initiated. The implementing institutions for three programmes (3, 10%) were not specified. About half (16, 53%) of the studies specified how lay vaccinators were selected: ten (10, 33%) reported selection by communities alone; selection was

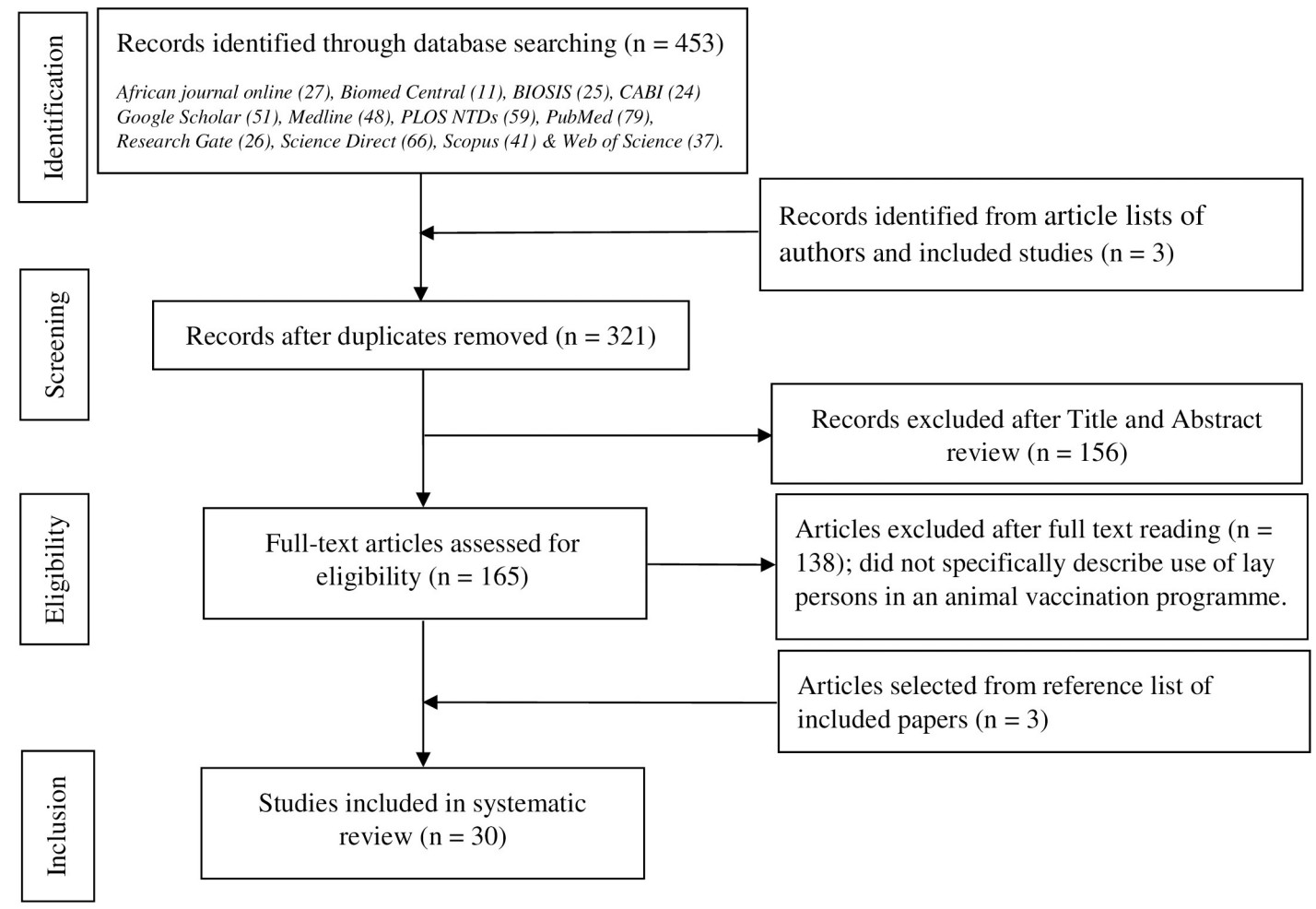

**Fig 1. Steps followed during selection of studies for inclusion in the review.**

done by both communities and programmes in three cases and in the other three cases, vaccinators were selected solely by programme officials following advertisement of the positions. Only seven out of 30 (23%) studies reported the length of the vaccinator training, which ranged from two to 28 days: three programmes trained up to three days, two training courses lasted from four to 21 days and two training courses took 22 days or more. About half (14, 47%) of the studies reported the content of the training delivered. The reported content included knowledge of disease transmission (7, 23%), vaccine administration and storage (10, 33%), farm management practices (1, 3%) and practical lessons that were undertaken to supplement the theory (6, 20%). The number of vaccinators were only reported by a few studies (6, 20%), four of which involved less than 36 vaccinators (Table 3).

The most frequently cited reasons for deployment of lay animal vaccinators were: limited professional veterinary services in under-resourced settings (12, 40%); remoteness which made accessibility to professional services difficult (8, 27%); lack of infrastructure to support services of professional veterinarians (6, 20%) and the inability of very small-scale farmers to afford the services of professional vets (6, 20%). Other reasons cited were frequent migration of nomadic pastoralists, who follow shifting pasture lands and rainfall patterns. Because the nomads' way of life posed pragmatic limitations on providing animal healthcare services using static, government or private set-ups, persons with traditional veterinary knowledge among

**Table 1. Overview of characteristics of reviewed studies.**

| Author | Year (Pub) | Country | Study setting | Objective of study | Study design | Data sources |
|---|---|---|---|---|---|---|
| Admassu [38] | 2005 | Ethiopia | Remote pastoral | Impact assessment and facilitating factors of change | Cross-sectional comparative | Semi-structured surveys through participatory approaches |
| Bagnol [39] | 2012 | Mozambique & Tanzania | Rural | Barriers to community involvement, from design to evaluation of an ND vaccination. | Cross-sectional, evaluation | IDIs, FGDs & programme reports |
| Belotto [15] | 1988 | Brazil | Urban & suburban | Organization and outcome of mass dog rabies vaccination | Cross-sectional, evaluation | Campaign & supervision reports |
| Bessell [40] | 2017 | Tanzania, India & Nepal | Rural | Uptake, outcome and impact of village-based ND vaccination | Before-after intervention assessment | Questionnaire survey and vaccination records |
| Brook [41] | 2010 | Canada | Remote | Delivery, needs and uptake of animal health services | Case study | Questionnaire survey and vaccination records |
| Bugeza [21] | 2017 | Uganda | Agro-pastoral | Performance of CAHWs | Participatory cross-sectional | Questionnaire, KIIs & FGDs |
| Cresswell [42] | 2014 | UK | Urban & rural pastoral | Uptake and usage of cattle vaccines | cross-sectional | Questionnaire survey |
| Curran & MacLehose [43] | 2002 | Low Income Countries | NR | Effects of CAHS on standard indicators for household wealth and health | Systematic review | Published, unpublished, in press and in progress |
| De Bryun [44] | 2017 | Tanzania | Rural | Uptake and outcomes of fee-for-service ND vaccination | Cross-sectional, evaluation | Questionnaire and programme records |
| Faris [45] | 2012 | Ethiopia | Remote, mainly pastoral | Seroprevalence and post-CAHWs-vaccination seroconversion rate of PPR with thermostable vaccine | Cross-sectional, evaluation | Lab and field questionnaire, & interviews |
| Harrison & Alders [46] | 2010 | Mozambique | Rural | Management practices and impact of ND vaccination programme | Cross-sectional survey | Questionnaire survey |
| Huttner [47] | 2001 | Malawi | Rural | Mortality, off-take and husbandry measures of user and non-users of CBAHS | Prospective cohort, evaluation | Questionnaire & farm record |
| Middaugh & Ritter [48] | 1982 | US (Alsaka) | Rural | Describe lay rabies vaccinator programme | Case study | Programme reports |
| Jones [49] | 1998 | Southern Sudan | Rural, agro-pastoral | Experiences of facilitating community-based animal health services | Programme Review | Programme & literature reports |
| Jost [22] | 1998 | Uganda | Remote pastoral | Economic, cultural and environmental information as indicators of impact of CAHW programme | Case control, evaluation | PRA, EIs, Lab & Field investigations |
| Kaare [50] | 2008 | Tanzania | Rural pastoral & agro-pastoral | Effectiveness of a variety of dog vaccination strategies | Randomized control trial | Household questionnaire & campaign records |
| Komba [51] | 2012 | Tanzania | Rural | Assess the efficacy of farmers delivering ND vaccines in village chickens | Experimental trial | Vaccination & serological records |
| Makundi [23] | 2012 | Tanzania | Rural pastoral | How marginalized pastoral communities are accessing animal health services | Cross-sectional | Semi-structured household questionnaire |
| Mariner [11] | 2012 | Several | Remote pastoral | Technical and institutional innovations leading to elimination of rinderpest | Programme review | Document & literature review |
| McCrindle [52] | 2007 | South Africa | Rural | Whether community volunteers could be trained to vaccinate village poultry | Experimental, evaluation | Participatory approaches, structured interviews, vaccination & serological records |
| Mgomezulu [53] | 2009 | Malawi | Rural | Efficacy and potency of I-2 ND vaccine administered by eye-drop in lab versus scavenging chickens | Experimental trial | Vaccination & serological records |
| Mockshell [54] | 2014 | Ghana | Suburban & Rural | Livestock keepers' perceptions of accessibility and affordability animal health service providers | Cross-sectional survey | Household survey & FGD |
| Mola [55] | 2018 | Ethiopia | Agro-pastoral | Effectiveness of CBAHWs in delivering primary animal health services | Post intervention survey | Questionnaire implementer survey |

(*Continued*)

**Table 1.** (Continued)

| Author | Year (Pub) | Country | Study setting | Objective of study | Study design | Data sources |
|---|---|---|---|---|---|---|
| Msoffe [56] | 2018 | Tanzania | Rural | Community involvement, context-specific and holistic animal health interventions | Intervention evaluation | Participatory approaches & vaccination records |
| Mugunieri [34] | 2004 | Kenya | Agro-pastoral | Nature, characteristics, and activities of CBAHWs | Programme Review | Observation, implementer & literature surveys |
| Mugunieri [35] | 2004b | Kenya | Agro-pastoral | Effectiveness of CBAHW programmes from the farmers' perspective | Evaluation | Ministry, project documents & literature review, & farmer questionnaire |
| Mwakapuja [57] | 2012 | Tanzania | Rural | Evaluation of immune status of free-range chickens inoculated by CVs | Randomized control trial | Vaccination & serological records |
| Nalitolela [58] | 2002 | Tanzania | Pastoral | Indicators and framework for monitoring impacts of CBAHW programme on household livelihood | Post intervention survey | Participatory approaches, semi-structured questionnaire |
| Schreuder [59] | 2014 | Afghanistan | Urban, suburban & rural | Impact assessment | Case comparative study | Survey, document & literature review |
| Swai [60] | 2014 | Tanzania | Pastoral | Characterize animal health delivery systems | Cross-sectional survey | Questionnaire, FGD, IDI and proportional piling |

CAHWs: Community Animal Health Workers CAHS: Community Animal Health Service CBAHWs: Community–Based Animal Health Workers CVs: Community Vaccinators EIs: Ethnoveterinary Interviews FGDs: Focus Group Discussions IDIs: In–depth Interviews KIIs: Key Informant Interviews ND: Newcastle Disease NR: Not Reported PRA: Participatory Rural Appraisals

the nomads were recruited by programmes and given some training to address animal health-care needs in their communities (6, 20%). Insecurity was also reported to have deterred professionals from certain environments and their roles were usually filled by lay animal health workers (5, 17%). A less commonly cited reason was mistrust of outsiders resulting in communities not readily accepting the services of outside professionals (1, 3%) (Table 3).

Diverse financial arrangements to cover the services of vaccinators included: i) salary-for-service, where a monthly salary or allowance was paid by the programmes (3, 10%); ii) fee-for-service, where vaccinators charged the full cost of their services, including profit to farmers (15, 50%); iii) cost recovery, where programmes gave materials and vaccines to vaccinators to service farmers without profit margins, part of the recouped costs were then returned to the programmes (as revolving funds) and the rest shared among involved stakeholders following agreed percentages (15, 50%); iv) service on credit or barter agreement, where vaccinators were paid by farmers with commodities other than money or had to wait to receive payment for their services later (1, 3%) and v) voluntary services, where the vaccinators received no remunerations and only derived motivation from the prestige associated with their work (5, 17%). Financial flexibilities likely made these programmes more amenable to implementation in poorer settings. Other operational strategies employed to facilitate implementation of the LAV programmes included supervision and monitoring by professional veterinarians (12, 40%), the opportunity for communities to participate in the delivery process including selection of vaccinators and monitoring of programmes (3, 10%), referral networks involving professional veterinarians (1, 3%) and certification of lay vaccinators (1, 3%) (Table 3).

## Extent of responsibilities of lay persons in animal vaccination programmes

Vaccinators vaccinated hooved livestock (cattle, camel, goat and sheep) against a range of diseases, including anthrax, blackleg, contagious bovine pleuropneumonia, contagious caprine

**Table 2. Summarized characteristics of studies reviewed.**

| Variables | Description | No. of studies [References] |
|---|---|---|
| Year study conducted | Before year 2000 | 4 [15,22,48,49] |
| | Year 2000–2010 | 11 [34,35,38,41,43,46,47,50,52,53,58] |
| | Year 2011–2020 | 15 [11,21,23,39,40,42,44,45,51,54–57,59,60] |
| Country of study | Afghanistan | 1 [59] |
| | Brazil | 1 [15] |
| | Canada | 1 [41] |
| | Ethiopia | 3 [38,45,55] |
| | Ghana | 1 [54] |
| | India | 1 [40] |
| | Kenya | 2 [34,35] |
| | Malawi | 2 [47,53] |
| | Mozambique | 2 [39,46] |
| | Nepal | 1 [40] |
| | South Africa | 1 [52] |
| | South Sudan | 1 [49] |
| | Uganda | 2 [21,22] |
| | United Kingdom | 1 [42] |
| | United States | 1 [48] |
| | Tanzania | 9 [23,39,40,44,50,51,57,58,60] |
| | Several countries | 2 [11,43] |
| Setting of study | Remote pastoral or agro-pastoral | 10 [11,21,22,34,35,38,41,45,55,58] |
| | Rural pastoral or agro-pastoral | 15 [23,39,40,44,46–53,56,57,60] |
| | Urban or Sub-urban | 4 [15,42,54,59] |
| | Unspecified | 1 [43] |
| Objective of study | Feasibility studies | 5 [45,48,50,52,53] |
| | Determinants of uptake of CAHW programmes | 11 [23,35,38–42,44,49,54,56] |
| | Outcome or impact assessment of CAHWs | 13 [11,15,21,22,34,38,40,43,44,47,55,58,59] |
| Study design | Case studies | 3 [41,48,59] |
| | Case control study | 1 [22] |
| | Randomized Control Trial or Experimental Studies | 5 [50–53,57] |
| | Cross-sectional surveys | 11 [15,21,23,38,39,42,44–46,54,60] |
| | Before–after intervention | 3 [40,55,58] |
| | Prospective cohort study | 1 [47] |
| | Systematic review/Review | 5 [11,34,35,43,49] |
| Data collection | Interviews | 6 [21,22,39,45,54,60] |
| | Laboratory reports (immunization outcomes) | 6 [22,45,51–53,57] |
| | Literature reviews | 6 [11,34,35,43,49,59] |
| | Ministry, programme or campaign reports | 17 [11,15,34,39–41,44,47–53,56,57,59] |
| | Non participant observations | 1 [34] |
| | Participatory/proportional piling approaches | 5 [22,52,56,58,60] |
| | Structured or semi-structured surveys | 18 [21,22,34,35,38,40–42,44–47,50,52,54,58–60] |

pleuropneumonia, peste des petits ruminants, hemorrhagic septicemia, lumpy skin disease, pasteurellosis and rinderpest (17, 57%); poultry against Newcastle Disease (ND) (12, 40%) and dogs against canine distemper virus, canine parvovirus and rabies (4, 13%). The types of vaccines used included: thermotolerant ND vaccine strains (10, 33%), Thermovax rinderpest vaccine (3, 10%), attenuated homologous PPR virus (Nigeria 75/1) strain vaccine (1, 3%), β-

**Table 3. Summary of the nature of use of Lay Animal Vaccinators (LAV).**

| Variables | Description | No. of studies [References] |
|---|---|---|
| Institution delivering LAV programme | Government initiatives | 5 [15,47,48,56,57] |
| | NGO projects | 12 [21,34,35,38–40,44,46,49,53,58,59] |
| | Research projects | 10 [22,23,41,42,45,46,50,52,54,55] |
| | Mixed or unspecified | 3 [11,43,60] |
| Year of project implementation | Before year 2000 | 11 [15,21,22,34,35,38,39,46,52,55,59] |
| | After year 2000 | 12 [39–41,44–46,50,52–54,56,57] |
| | Unspecified | 7 [11,23,42,43,51,55,60] |
| Mode of selection of LAVs | By communities alone | 10 [21,22,39,46,47,49,51,52,55,56] |
| | By communities & programme | 3 [15,41,44] |
| | By programme alone | 3 [40,53,57] |
| Content of training | Knowledge of disease transmission | 7 [15,38,44,49,53,55,56] |
| | Vaccine administration and storage | 10 [15,40,44,46,48–50,53,57,60] |
| | Farm management practices | 3 [53,56,60] |
| | Added practical sessions | 6 [42,47,50,53,55,56] |
| Length of training | ≤ 3 days | 3 [15,39,40] |
| | 4–21 days | 2 [49,55] |
| | 22 or more days | 2 [59,60] |
| Number of LAVs used/studied | ≤ 100 | 6 [38,39,51,52,55,60] |
| | > 1,000 | 1 [59] |
| Reasons for using lay animal vaccinators | Limited professional service | 12 [23,35,38,41,45,47–49,51,54,55,60] |
| | Remoteness | 8 [21,22,35,38,41,45,49,60] |
| | Infrastructure | 6 [11,22,35,40,41,49] |
| | Affordability | 6 [21,35,40,41,49,50] |
| | Nomadic | 6 [21–23,45,50,58] |
| | Insecurity | 5 [11,21,22,49,59] |
| | Mistrust | 1 [41] |
| Remuneration arrangements | Cost recovery | 16 [11,21,22,34,35,38,40,42–44,49,53,55,57,59,60] |
| | Fee-for-service | 15 [21,23,34,35,38–40,42–44,46,51,54,58,59] |
| | Voluntary service | 5 [15,45,48,52,56] |
| | Salary for service | 3 [50,55,59] |
| | Service on credit or barter agreement | 1 [49] |
| Operational strategies | Supervision/monitoring by professional vets | 12 [11,34,35,39,45,47,49,55,56,58–60] |
| | Community participation | 5 [15,39,46,51,56] |
| | Referral systems | 2 [21,60] |
| | Certification of LAVs | 1 [48] |

propiolactone activated suckling mouse brain vaccine for rabies (1, 3%), Nobivac Rabies & Puppy DP vaccines (against rabies, canine distemper virus and canine parvovirus) (1, 3%), and unspecified rabies vaccine (2, 6%). These studies also reported that vaccinators administered other injectable formulations to treat livestock diseases, including antibiotics (e.g. tetracyclines), trypanocidal and anthelmintic drugs (13, 43%). Apart from vaccinating, lay animal vaccinators were also reported to have played other roles, including encouragement of responsible dog ownership and management as part of dog population control measures (1, 3%), sensitization and awareness raising of animal health programmes (2, 6%), and as general advisors to farmers regarding animal health (1, 3%) (Table 4).

**Table 4. Summary of the extent of use of Lay Animal Vaccinators (LAV).**

| Variables | Description | No. of studies [References] |
|---|---|---|
| Roles played by LAVs | As vaccinators | 30 All studies |
| | As agents of sensitization | 1 [40,41] |
| | As advisors of farmers | 1 [47] |
| | Dog population control | 1 [41] |
| Biologics delivered | Multiple vaccines for livestock diseases | 13 [21,23,34,35,38,42,43,47,54,55,58–60] |
| | Thermotolerant ND vaccine strains | 10 [39,40,43,44,46,51–53,56,57] |
| | Thermovax rinderpest vaccine | 3 [11,22,49] |
| | Rabies vaccine (unspecified) | 2 [41,48] |
| | Nobivac Rabies & Puppy DP vaccines | 1 [50] |
| | β-propiolactone inactivated suckling mouse brain vaccine for rabies | 1 [15] |
| | Attenuated homologous PPR virus (Nigeria 75/1) strain vaccine | 1 [45] |
| Animals vaccinated | Camel, Cattle, Goat, Sheep | 17 [11,21–23,34,35,38,42,43,45,47,49,54,55,58–60] |
| | Poultry | 12 [39,40,43,44,46,47,49,51–53,56,57] |
| | Dogs | 4 [15,41,48,50] |

## Effectiveness and other benefits of lay animal vaccinator programmes

This review generally found positive outcomes reported from lay animal vaccinator programmes in the contexts where they were used. Close to half of the studies (13, 43%), which included case, case-control, randomized control or experimental trials, prospective cohort or before-after studies reported increased flock and herd sizes and improved farmer knowledge of best farm management practice, with contributions to improved livelihoods and farmer assurance in animal assets. Some studies also reported decreased disease transmission and outbreaks, reduced mortality among vaccinated animal populations and for some zoonotic diseases (11, 37%). Several studies reported high vaccination coverage, which in some cases were superior to those achieved by professional-led programmes (10, 33%). In studies that compared sero-conversion and birth rates among herds vaccinated by lay vaccinators versus those vaccinated by professional veterinarians, no significant differences were reported (3, 10%) (Table 5).

**Table 5. Effectiveness and facilitating factors of success of lay animal vaccinator programmes.**

| Variables | Description | No. of studies [References] |
|---|---|---|
| Effectiveness and other benefits of programmes | Increased farm output and farmer assurance in animal assets | 13 [11,15,34,35,39,40,44,46,47,49,50,53,58] |
| | Decreased disease transmission and case mortality | 11 [15,22,38,46,47,49,51,55,57–59] |
| | High vaccination coverage | 10 [11,15,39,40,44,49–51] |
| | Non-inferior in birth and immunization outcomes | 3 [35,45,49] |
| Factors facilitating success of programmes | Community participation | 14 [11,15,22,39,44–46,49,51,52,55,56,58,59] |
| | Public support | 13 [11,15,22,35,39,41,42,48,49,54,58–60] |
| | Training | 9 [22,35,45,48–50,56,58,59] |
| | Ethnoveterinary knowledge | 6 [11,22,23,38,49,58] |
| | Compensation | 5 [39,49,50,58,59] |
| | Awareness | 5 [15,40,44,56,58] |
| | Affordability | 5 [15,22,52,54,58] |
| | Thermostable vaccine | 5 [35,42,54,58,61] |
| | Accessibility | 4 [11,49,52,53] |
| | Flexibility and familiarity with terrain | 4 [11,22,52,60] |

None of the 30 studies reviewed reported cost-effectiveness of the lay vaccinator programmes they assessed as it did not form part of their study objectives.

## Facilitating factors, challenges and current steps towards making lay animal vaccination programmes mainstream

The most frequently cited contributory factor to the success of lay animal vaccinator programmes was the opportunity for communities to participate in the selection of vaccinators, and in the delivery and monitoring of the interventions (14, 47%). The second and third most important factors were public support (13, 43%) and comprehensive training for all stakeholders, with programmes that trained community leaders and farmers in addition to implementers recording more positive outcomes (9, 30%). Other important facilitating factors were the level of ethnoveterinary knowledge shown by vaccinators which aided their training (6, 20%), satisfactory compensation for the vaccinators (5, 17%), heightened awareness of the programmes amongst beneficiary livestock keepers (5, 17%) and relative affordability of the services provided (5, 17%). The advent of thermostable vaccines enabled lay persons with minimal training to handle and administer the biologics under indirect technical supervision by professional veterinarians (5, 17%). The lay animal vaccinators were also cited to be relatively more accessible (4, 13%), more trusted, more flexible, and more familiar with local terrains compared to professional veterinarians, which enabled them to deliver the interventions in varying socioeconomic contexts (4, 13%) (Table 5).

Eighteen of the 30 studies (60%) reported challenges faced by these programmes. The most frequently reported challenges were: lack of legalized institutional support for the programmes (8, 27%), competing financial interests of vaccinators and community leaders which derailed success of programmes, for instance in some programmes, community leaders criticized financial arrangements where vaccinators autonomously collected fees from farmers for their services; village leaders complained that the vaccinators went to training and received per diems, T-shirts and caps; village authorities placed a tax on birds vaccinated in a programme which was viewed negatively by farmers; vaccinators who received remuneration per bird vaccinated tended to focus only on households with larger flock sizes; also, some vaccinators left the project once they found paying work inside or outside of the village (8, 27%) and lack of comprehensive programming and communication of benefits led to declining participation by farmers (8, 27%). There also were occasions where the performance of lay vaccinators was poor and concerns arose about their level of professionalism, technical capabilities and effectiveness (6, 20%). In a few cases, there was inadequate engagement between implementing organizations and local professional veterinarians, eliciting opposition to the programmes (3, 10%). Sometimes farmers mistrusted and did not readily accept the services of lay vaccinators,

**Table 6. Frequency of themes relating to challenges faced by lay vaccinator programmes.**

| Variables | Description | No. of studies [References] |
|---|---|---|
| Challenges faced | Lack of institutional support | 8 [11,15,21,34,35,39,49,58] |
| | Competing interests of stakeholders | 8 [15,34,35,39,43,44,49,52] |
| | Programming issues | 8 [21,34,35,39,49,52,53,56] |
| | Limited capacity of vaccinators | 6 [21,38,42,45,52,54] |
| | Inadequate stakeholder engagement | 3 [21,39,49] |
| | Mistrust by communities | 3 [21,41,49] |
| | Limited resourcing | 2 [21,49] |
| | Insecurity | 1 [49] |

for example fearing they were providing information about their herds to the government for the purpose of taxation (3, 10%). Finally, lay vaccinators were sometimes poorly resourced (2, 6%) and also affected by the insecurity (unrest) in communities (1, 3%) (Table 6).

The review did find evidence of intense opposition to use of lay animal vaccinators from professional veterinarians, mostly citing concerns of low levels of professionalism and technical competence [11,34,35,58]. We found no record of lay vaccinator programmes being integrated into national veterinary services, even in countries with large, remote, and underserved populations that are highly dependent on livestock for livelihoods and food security. In this regard, this review did not find a single regulatory framework for lay vaccinator programmes from any of the low and middle-income countries, however, a few studies described policy instruments that prescribe selection processes, minimum training, certification and regulation of lay animal vaccinator activities in high income countries including Canada, UK and the USA [41,42,48,54]. In response to these challenges, 13 (43%) of the 30 studies reported proposed or implemented solutions, which broadly included: training, supervision, legal and policy backing, inclusive incentive regimes and greater involvement of communities in the planning and execution of these programmes (Table 7).

## Discussion

This scoping review identified extensive use of lay persons in the implementation of animal vaccination programmes designed for various purposes including research studies, vaccination campaigns and continuous provision of animal health services to farmers in poor and hard-to-reach communities. The review indicates that studies demonstrating benefits and positive perceptions of deployment of lay animal vaccinators outnumbered those identifying problems and challenges. As such, development of training curricula and regulatory policy frameworks should be considered to improve the quality, uptake and benefits of such programmes. This will likely promote high-quality practices among lay animal vaccinators, generate public support for their services and allow for better integration and recognition within animal health services.

The review showed that lay animal vaccinators generally performed well in delivering animal vaccinations to their communities. In most cases, they achieved similar or better outcomes in comparison with professional veterinarian counterparts [22,45,49]. This suggests that if

**Table 7. Proposed and implemented solutions to challenges faces by the lay animal vaccinator programmes.**

| Variables | Description | No. of studies [References] |
|---|---|---|
| Proposed or implemented solutions to challenges | Refresher training; minimum training requirement; training all local stakeholders | 5 [21,34,38,54,56] |
| | Policy backing for lay animal vaccinator programmes | 3 [34,56,60] |
| | Gradual replacement or vertical progression of lay animal vaccinators | 2 [21,34] |
| | Supervision and information sharing/ technical support | 2 [42,60] |
| | Combined vaccines to target multiple endemic diseases (e.g. sheep pox & PPR) or integrated interventions | 2 [34,45] |
| | Community leaders put in control | 1 [39] |
| | Engage locals, develop culturally relevant methods | 1 [41] |
| | Functionally regulated drug market | 1 [49] |
| | Provide mutually favorable incentives for communities and professionals | 1 [11] |
| | Institute salary or fee for service | 1 [52] |
| | Tailor intervention to local needs | 1 [56] |

trained and resourced, lay vaccinators can provide valuable contributions to the prevention and control of animal diseases amenable to mass vaccination. In this regard, vaccinators who received regular training were engaged more often than those that did not [35]. However, these findings should be used with consideration of the possibility that studies demonstrating positive outcomes may be more likely to be published than those demonstrating negative outcomes. Table 1 also provides information on the methodologies of included studies and can be useful in interpreting the findings of this review.

The review also noted that, following the move to privatize veterinary services, private veterinarians have been hesitant to establish themselves in rural and remote communities where infrastructure to support the veterinary practice is limited and most farming is carried out on a subsistence basis, with subsistence farmers having limited purchasing power [20,23]. Conflict and harsh climatic conditions may also have been contributory factors to reduced trade volumes in some of these areas through reduced cash circulation [49]. This is consistent with the finding that, despite demand for veterinary services, the most common reason for establishing lay vaccinator schemes was limited availability of government or private professional veterinarians in remote and rural settings.

Another contributory factor to the rising interest in use of lay animal vaccinators is the discovery or development of thermostable properties of some animal vaccines, which has opened up possibilities for maintaining year-round animal vaccination in remote communities with the potential of sustaining high vaccination coverage. By way of example, the Nobivac rabies vaccine was shown in 2016 to maintain its potency when stored at 25°C for up to six months [33]. However, as observed in the case of Thermovax Rinderpest vaccine, it can take many years to take advantage of such developments [11]. Therefore, the thermostable properties of vaccines for important animal diseases like peste des petits ruminants (PPR) and rabies may not be exploited for a long time if policy makers are not fully engaged with available evidence or do not recognize the implications for implementing such animal vaccination programmes.

While many positive outcomes of lay animal vaccinator programmes were identified through this review, several of the studies also reported intense opposition to the initiatives by established veterinary systems and absence of policy frameworks to regulate their deployment. Notably from this review, we did not find a single regulatory policy for lay animal vaccinators programmes from any developing countries where the services of lay vaccinators are arguably most needed. The major fears of professionals relate to suboptimal performance and the possibility that persons trained only to inoculate animals on one programme will begin to hold themselves up as veterinarians, proving services to farmers beyond their training and potentially cause harm [21,38,42,52].

Consequently, a key step towards advancing lay animal vaccination lies in addressing the barriers to acceptance. A comprehensive stakeholder engagement could bring together groups such as implementers, researchers and policy makers to discuss the potential of such programmes and concerns over their deployment. The provision of institutional support was a critical factor to positive outcomes of the lay vaccinator programmes and is supported by the finding that it was among the most cited challenge as well as facilitating factor to implementation. The second most cited facilitating factor related to public support, indicating that the lay-vaccinator-programmes were most effective where they were underpinned by legal frameworks and community support. Even though established systems are central to discussions on formalizing lay animal vaccination services, other institutions such as researchers, farmer groups and non-governmental agro-institutions have key roles in ensuring such discussions are sustained to create programmes that meet the needs of communities. However, implementing such lay animal health worker programmes across the rural landscape of low- and middle-income countries can be challenging in several ways, for example, it may not be

feasible for trainees to adequately attain the required competences due to logistical constrains and social factors. Monitoring and enforcing regulations could also risk overwhelming veterinary systems that are already burdened and inadequately resourced.

The review also noted enhanced acceptability and outcomes of lay animal vaccinator programmes that involved substantial community participation [62,63]. However, given that the majority of schemes were instituted and funded through external development or research programmes, with little or no local inputs, their continuation may have been problematic [20,43,44,52]. A holistic approach to the design of programmes, taking into account the needs of communities regarding common local diseases, and clearly communicating programme objectives and benefits are more likely to generate and sustain stakeholder interests. For instance, if a Newcastle disease vaccination programme is portrayed as if it will (unrealistically) prevent all poultry deaths rather than its more realistic outcome of reducing the risk of deaths among flocks, then outbreaks from other poultry diseases will likely cause distrust and loss of confidence in the programme [52,53,64].

Other major concerns fueling resistance to lay animal vaccination programmes relate to their level of professionalism and technical competence. The argument is made that lay vaccinators could potentially do harm, for example in attempting to sell drugs that farmers do not need, hiking prices or diluting drugs [25]. However, researchers have reported contrasting outcomes, with lay animal vaccinators providing an alternative to untrained drug vendors and also transferring knowledge to farmers [34,35]. To therefore make the case for lay animal vaccination programmes, implementing institutions must present comprehensive data as to how long and how much training is adequate for optimal performance. A formalized training guide such as that designed for paraprofessional veterinarians by the OIE would be useful in establishing and disseminating widely accepted standards for training vaccinators [16]. Also, national programmes that allow lay animal vaccinators with requisite qualifications to progress vertically can further ensure continuity and gradually increase the numbers in higher cadres of veterinary professionals. To this end, we recommend that future studies that aim to assess outcomes and impacts of lay animal vaccinator programmes should also report the duration and content of trainings given to the implementers, to guide discussions of their performance and steps to improve it.

The use of non-participant observation approaches was rarely employed in the included studies but could be useful in capturing information on vaccinator professional conduct and competence. It would also be useful to conduct larger scale randomized control trials on lay animal vaccinator programmes. This will enhance the validity and generalizability of findings, as most of the included studies in this review were small scale; only six of included studies reported numbers of lay vaccinators used or studied and in those reports the number of lay vaccinators were less than 100. Again, although one of our objectives was to review the cost effectiveness of lay animal vaccinator programmes there was no data on this and as such future studies should aim to capture more information on the impact and cost-effectiveness to assist policy formulation for the sector. However, since the lay animal vaccinator programmes involved some level of volunteering, this may have resulted in the interventions operating at reduced costs [15,45,48,52,56,65].

Finally, the adoption of lay animal vaccinators is analogous to the shifts of some medical tasks in human healthcare, where nurses and health assistants are permitted to perform less technical care to afford physicians time to attend to more complex cases. This concept is widely being applied in clinical settings and community-level public health interventions, leading to better efficiency and cost-savings [1,2,4,66–70]. Similarly, national regulations that stipulate the processes leading to attainment of technical competence and professionalism for lay animal vaccinators would help to generate institutional and farmer support for their services

[14,41,48,71]. Indeed, the implementation of mass vaccination against Covid-19, which will require rapid mobilization of a very large workforce, will undoubtedly bring the discussion of lay vaccinators into sharp focus and further lessons may well be learned that have relevance for implementation of mass animal vaccination campaigns.

Our systematic approach to the literature search generated useful insights in assessing the nature and extent of deployment of lay animal vaccinators, but did have some limitations. Several studies were excluded from this review because the qualification of the vaccinators was not reported and we did not obtain further information after contacting the study authors. The strong opposition to use of lay animal vaccinators may have discouraged researchers from reporting vaccinator qualifications or even led to researchers not publishing such works at all. Additionally, our sampling may have been skewed toward studies that recorded positive outcomes of lay animal vaccinators programmes as it appears these are more likely to be published than those that recorded negative outcomes. We also did not perform a quality appraisal (rating of the methodological soundness) of included studies, which could have informed readers of the validity and reliability of study findings. Quality appraisal is usually not a necessary requirement for a scoping review. We also did not include studies published in languages other than English and therefore may have missed out on relevant studies.

## Conclusions

We have demonstrated here that lay animal vaccinator schemes have been appreciably deployed to prevent a wide range of animal diseases by research institutions, government and non-government developmental organizations and in many cases achieved positive animal and public health outcomes. Despite their potential to ameliorate the challenges posed by limited availability of professional veterinarians and to support roll out of mass animal vaccination campaigns in resource-constrained settings, countries have not taken steps to integrate their services into mainstream veterinary systems. Policy frameworks that prescribe how lay animal vaccinators are selected, trained, certified, deployed and monitored, and co-designing programmes with local communities so they are tailored to their needs, will go a long way to generating public support and confidence in these services. However, these have to be done alongside putting in place robust monitoring and enforcement systems to ensure safe and quality animal health care delivery.

## Author Contributions

**Conceptualization:** Christian Tetteh Duamor, Katie Hampson, Felix Lankester, Sally Wyke, Sarah Cleaveland.

**Data curation:** Christian Tetteh Duamor, Maganga Sambo, Sarah Cleaveland.

**Formal analysis:** Christian Tetteh Duamor, Katie Hampson, Felix Lankester, Maganga Sambo, Katharina Kreppel, Sally Wyke, Sarah Cleaveland.

**Funding acquisition:** Katie Hampson, Felix Lankester, Katharina Kreppel, Sally Wyke, Sarah Cleaveland.

**Investigation:** Christian Tetteh Duamor, Katie Hampson, Maganga Sambo, Sarah Cleaveland.

**Methodology:** Christian Tetteh Duamor, Katie Hampson, Felix Lankester, Sally Wyke, Sarah Cleaveland.

**Project administration:** Christian Tetteh Duamor, Katie Hampson, Felix Lankester, Sally Wyke, Sarah Cleaveland.

 

**Supervision:** Christian Tetteh Duamor, Katie Hampson, Felix Lankester, Sally Wyke, Sarah Cleaveland.

**Validation:** Christian Tetteh Duamor, Katie Hampson, Felix Lankester, Maganga Sambo, Katharina Kreppel, Sally Wyke, Sarah Cleaveland.

**Visualization:** Christian Tetteh Duamor, Katie Hampson, Sarah Cleaveland.

**Writing – original draft:** Christian Tetteh Duamor.

**Writing – review & editing:** Christian Tetteh Duamor, Katie Hampson, Felix Lankester, Maganga Sambo, Katharina Kreppel, Sally Wyke, Sarah Cleaveland.

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
