## [Decision Letter · Decision Letter 0]

30 Apr 2021

Dear Mr. DUAMOR,

Thank you very much for submitting your manuscript "Use of lay vaccinators in animal vaccination programmes: A scoping review." for consideration at PLOS Neglected Tropical Diseases. As with all papers reviewed by the journal, your manuscript was reviewed by members of the editorial board and by several independent reviewers. The reviewers appreciated the attention to an important topic. Based on the reviews, we are likely to accept this manuscript for publication, providing that you modify the manuscript according to the review recommendations. 

We received three constructive reviews of this manuscript and all of the reviewers were generally positive about the work. The reviewers highlight issues with regard to potential bias in article selection and findings that need to be addressed.

Sincerely,

Amy J Davis, Ph.D.

Associate Editor

Amy Gilbert

Deputy Editor

I have received three constructive reviews of this manuscript. All reviewers were generally positive about the work. The reviewers highlight issues with regard to potential bias in article selection and findings that need to be addressed.

Reviewer's Responses to Questions

**Key Review Criteria Required for Acceptance?**

**Methods**

-Are the objectives of the study clearly articulated with a clear testable hypothesis stated?

-Is the study design appropriate to address the stated objectives?

-Is the population clearly described and appropriate for the hypothesis being tested?

-Is the sample size sufficient to ensure adequate power to address the hypothesis being tested?

-Were correct statistical analysis used to support conclusions?

-Are there concerns about ethical or regulatory requirements being met?

Reviewer #1: I worry that the search criteria used to identify studies was too narrow (Line 161). I recognise that highlighting individual publications as examples in such an exercise is unhelpful where they may have been omitted in the review methods described, but from the rabies literature alone the following recent paper would appear to not have met the initial search criteria stated in Line 161 due to not having used the terms “vaccinator” or “animal health worker”, despite appearing to align with the scope of the review: “Volunteer based approach to dog vaccination campaigns to eliminate human rabies: Lessons from Laikipia County, Kenya” (https://doi.org/10.1371/journal.pntd.0008260). This paper may have fallen out in the subsequent screening, but my concern is that if it were not captured in the initial search criteria then many relevant publications may have been missed. Apologies if I have misinterpreted the current search methodology and papers such as this were captured and excluded.

Otherwise the methods appear appropriate and robust.

Reviewer #2: My main concern with the paper (albeit a small concern) is that the sampling method can be expected to have a strong bias towards programs that were successful. The authors admit this selectively in lines 394-396 and 485. I therefore find it hard to trust the quantitative conclusions drawn (not only the statement preceding line 394). 

I propose including a general statement on the limitations of the sampling method and consequently all the reported proportions.

Reviewer #3: (No Response)

**Results**

-Does the analysis presented match the analysis plan?

-Are the results clearly and completely presented?

-Are the figures (Tables, Images) of sufficient quality for clarity?

Reviewer #1: The results are well presented in the text, with clear and easy to understand tables.

Reviewer #2: Some of the headings of Figure 1 have been cropped (Identificatio_, Eligibili__).

Reviewer #3: (No Response)

**Conclusions**

-Are the conclusions supported by the data presented?

-Are the limitations of analysis clearly described?

-Do the authors discuss how these data can be helpful to advance our understanding of the topic under study?

-Is public health relevance addressed?

Reviewer #1: The discussion is well structured and largely well balanced, highlighting the benefits and challenges around the use of lay vaccinators. Clearly there is a huge need to to expand the available workforce for vaccine delivery across LMICs to improve human and animal health, however there is one aspect which I feel needs further consideration in the discussion.

Concerns around lay vaccinators potentially doing harm is mentioned in the discussion (Line 449), in the misuse of drugs, however the possibility of over-reach in their clinical duties is not discussed. This is alluded to as a finding from the literature in Line 313, where it states they take part in dog population control activities and sterilisation, presumably meaning lay persons performing surgical procedures. It would be important to flag the considerable animal welfare issues resulting from untrained and poorly equipped persons undertaking certain veterinary duties and the challenges in monitoring such practices in the settings in question. The need for regulation is well discussed, but the perspective of devolving veterinary responsibilities without enforceable regulation is not highlighted, along with the potential for this to undermine the sometimes frail existing veterinary frameworks established to date. Largely these concerns are reflected in the authors' emphasis on the need for involvement of the veterinary sector, training, regulation and strong OIE guidelines etc, however inclusion of the risks of devolution without successful regulation would be an important addition. Examples are given of where lay vaccinators have been used in the developed world to rapidly expand the vaccinator workforce, however well-established regulatory frameworks have enabled this to be done in a controlled manner. Applying the same principles in rural areas of LMICs carries vastly different challenges in communicating and enforcing such regulations and I feel that this should be given greater weight in the discussion and conclusion.

Reviewer #2: No comments.

Reviewer #3: (No Response)

**Editorial and Data Presentation Modifications?**

Reviewer #1: (No Response)

Reviewer #2: No comments.

Reviewer #3: (No Response)

**Summary and General Comments**

Reviewer #1: The study provides a systematic review of the literature in the benefits and limitations to the use of lay vaccinators in rabies control campaigns. Congratulations to the authors for this timely and relevant review which provides a concise summary of a broad range of applications of lay vaccinators. The review stimulates innovative thinking in how to tackle the issue of access to veterinary services across many LMICs.

Reviewer #2: In my view the paper has something to offer to the scientific community, namely that lay people present an underutilized resource.

I think that the paper could benefit from the inclusion of some evidence from outside of the scientific literature (consider personal communications or press reports). This could strengthen some of the stated arguments.

Reviewer #3: (No Response)

PLOS authors have the option to publish the peer review history of their article (what does this mean?). If published, this will include your full peer review and any attached files.

Reviewer #1: No

Reviewer #2: Yes: JL Kotze

Reviewer #3: No

Figure Files:

Data Requirements:

Reproducibility:

References

---

## [Editor Report · Decision Letter 1]

28 Jul 2021

Dear Mr. DUAMOR,

We are pleased to inform you that your manuscript 'Use of lay vaccinators in animal vaccination programmes: A scoping review.' has been provisionally accepted for publication in PLOS Neglected Tropical Diseases.

Best regards,

Amy J Davis, Ph.D.

Associate Editor

Amy Gilbert

Deputy Editor

Lines 43-45: Rephrase this sentence for clarity. Reported positive outcomes of what? And how do lay vaccinators lead to larger flock and herd sizes?

Lines 47-49: “Facilitating factor” in unclear, and not clear how financial interests of stakeholders is a barrier. Reword.

Avoid using pronouns such as ‘they’ or ‘it’ unless ‘they’ or ‘it’ has already been described in that sentence (examples, lines 60 and 63).

<style type="text/css">p.p1 {margin: 0.0px 0.0px 0.0px 0.0px; line-height: 16.0px; font: 14.0px Arial; color: #323333; -webkit-text-stroke: #323333}span.s1 {font-kerning: none

</style>

---

## [Editor Report · Acceptance letter]

5 Aug 2021

Dear Mr. Duamor,

We are delighted to inform you that your manuscript, "Use of lay vaccinators in animal vaccination programmes: A scoping review.," has been formally accepted for publication in PLOS Neglected Tropical Diseases.

Best regards,

Shaden Kamhawi

co-Editor-in-Chief

Paul Brindley

co-Editor-in-Chief
